# MLGLP: Multi-Scale line-graph Link Prediction based on Graph Neural Networks

## Abstract

This manuscript proposes a multi-scale link prediction approach based on Graph Neural Networks (GNNs). The proposed method - Multi-Scale Line-Graph Link Prediction (MLGLP) - learns the graph structure and extracts effective representative features of graph edges to address challenges such as information loss and handle multi-scale information. This approach utilizes embedding vectors generated by GNNs from enclosing subgraphs. While expanding GNN layers can capture more intricate relations, it often leads to overs-smoothing. To mitigate this issue, we propose constructing coarse-grained graphs at three distinct scales to uncover complex relations. To apply multi-scale subgraphs in GNNs without using pooling layers that lead to information loss, we convert each subgraph into a line-graph and reformulate the task as a node classification problem. The hierarchical structure facilitates exploration across various levels of abstraction, fostering deeper comprehension of the relationships and dependencies inherent within the graph. The proposed method is applied on link prediction problem, which can be modelled as a graph classification problem. We perform extensive experiments on several well-known benchmarks and compare the results with state-of-the-art link prediction methods. The experimental results demonstrate the superiority of our proposed model in terms of average precision and area under the curve.

## 1 INTRODUCTION

In our increasingly interconnected world, many problems can be seen as graph-structured data. Graphs are mathematical structures composed of nodes (vertices) and edges (links or connections between nodes). They are widely used to represent complex systems in various domains such as social networks, biological networks, recommendation systems, chemistry, citation networks, and power networks Cai et al. (2021). Link prediction is a fundamental task in graph analytics with diverse applications across multiple domains such as friend recommendations in social networks or predicting interactions between genes Zhu et al. (2023). Traditional methods for link prediction use graph structural properties (e.g., common neighbors), often overlook important information such as features associated with each node. In contrast, graph representation-learning techniques such as Graph Neural Networks (GNNs), integrate graph structures with node/edge features, allowing the capture of complex relationships through iterative message passing across graph edges.

GNN-based link prediction methods are categorized into node-based and subgraph-based approaches Zhang et al. (2020). Node-based techniques like Graph Convolutional Networks (GCN) Yao et al. (2019), Graph Attention Network (GAT) Veličković et al. (2017), GraphSAGE (SAGE) Hamilton et al. (2017), and Graph AutoEncoder (GAE) Kipf & Welling (2016) incorporate multi-hop graph structures through message passing. They first extract the node embeddings and then predict the possible link between two nodes using a similarity method or a classifier, such as a multi-layer perceptron, to process both representations and determine the likelihood of a link between them. In contrast, subgraph-based methods, such as SEAL Zhang & Chen (2018), BUDDY Chamberlain et al. (2022), mLink Cai & Ji (2020), LGLP Cai et al. (2021), and LGCL Zhang et al. (2023) extract an h-hop enclosing subgraph around the target link, learning a representation specifically tailored to that subgraph. They then use a binary classifier to determine whether the subgraph indicates the presence or absence of a link Li et al. (2024). However, methods like SEAL may struggle when node representations closely resemble those of their local neighborhoods, as they may fail to effectively capture information from distant neighbors. This limitation results in subgraphs

containing redundant or irrelevant details, which can diminish model performance. If nodes share similar features with nearby nodes, they may miss crucial information from farther nodes, leading to less accurate predictions. A possible solution is to expand the GNN layers to extract more intricate relations. However, it faces challenges like over-smoothing, where nuanced messaging becomes difficult, resulting in node homogenization, particularly in limited subgraph scenarios. Multi-scale methods such as mLink Cai & Ji (2020) provide hierarchical information retrieval capabilities. However, they rely on pooling layers to handle varying node numbers in subgraphs, which may result in information loss.

**Present work**. To address these challenges, in this paper we propose a link prediction approach that seamlessly integrates both local and global graph structures into a unified framework that allows for a more complete understanding of the graph's structure. The main idea is to learn relations and features from subgraphs extracted at different scales—ranging from local (small) to global (large) subgraphs. Each scale captures information at a different level of granularity. This allows the model to understand relationships between nodes at multiple levels of detail. Instead of directly predicting links between two nodes, we reformulate the link prediction task as a binary node classification problem. To achieve this, we first convert the original graph into a line-graph, where each node corresponds to an edge in the original graph. Since each node in the line-graph represents a link in the original graph, a classification of 1 implies the existence of a link between two nodes in the original graph, while a classification of 0 indicates no link. We apply a GCN on the line graph at each scale to obtain node embeddings. For each node, the embeddings generated at different scales are concatenated and fed into a multi-layer perceptron (MLP) to predict the node's class. Hierarchical structures resulting from different graph scales, enable the analysis of graphs at different levels of granularity, allowing us to group nodes based on their relationships and capture broader patterns. By considering not only individual nodes but also their collective interactions within subgraphs or clusters, our method extracts richer information and provide a more nuanced understanding of the graph structure for classification purposes. The results obtained from experiments on real-world datasets demonstrate the superiority of the proposed method compared to state-of-the-art approaches. The main properties of the proposed method listed as follows:

1. Our method incorporates both local and global graph structures by leveraging subgraphs at different scales, allowing for a more comprehensive understanding of graph relationships, capturing both micro and macro-level graph information.

2. Instead of directly predicting links, our approach reformulates the link prediction problem as a node classification task using line-graphs. This approach helps in reducing information loss and simplifying the learning process for the model.

3. The approach considers individual node attributes and their collective interactions within subgraphs, enabling richer feature extraction and a more nuanced representation of the graph's structure and relationships.

## 2 RELATED WORKS

Several approaches have been proposed to address the link prediction task, which can be classified into proximity-based (non-learning) and learning-based approaches. Proximity-based methods rely on statistical properties of nodes/edges within the graph without explicitly learning embedding of nodes or edges. They typically use heuristic or feature-based techniques. Heuristic methods generally employ predefined rules or measures and evaluate link existence by assigning scores derived from the graph structure, utilizing either common neighbors or path information. Examples include Common Neighbour (CN) Newman (2001), Adamic Adar (AA) Adamic & Adar (2003), Resource Allocation (RA) Zhou et al. (2009), Significant Influence (SI) Yang et al. (2018), Shortest Path, and Katz Katz (1953). They often assess the existence of a link by assigning a score derived from the graph structure. CN, AA, and RA methods primarily depend on common neighbors, whereas SI, Shortest Path, and Katz methods utilize the graph paths. These methods are widely used due to their simplicity and interoperability. However, each heuristic relies heavily on an underlying assumption regarding the likelihood of two nodes forming a connection, which constrains their efficacy when these assumptions are not met in certain network contexts. Moreover, they rely solely on graph structure and overlook node or edge features, often effective in link prediction tasks. Feature-based methods, on the other hand, use machine learning models trained on a set of features, such as node,

edge, or graph attributes. While they incorporate explicit features, they may not fully exploit the underlying graph structure, potentially missing important relational information and dependencies. This can lead to less accurate or insightful models than those that utilize the graph structure directly.

Representation learning methods transform the graph structure into a low-dimensional vector space. They are divided into embedding-based and GNN-based methods. Popular embedding-based methods include Matrix Factorization (MF) Menon & Elkan (2011), MLP, Large-scale Information Network Embedding (LINE) Tang et al. (2015), DeepWalk Perozzi et al. (2014), and node2vec Qiu et al. (2018). For instance, DeepWalk Perozzi et al. (2014) uses the random walk strategy to generate node sequences and applies the Skip-gram model to learn node embeddings. node2vec Qiu et al. (2018) method extends the DeepWalk by using a biased random walk to better explore neighborhoods. The LINE Tang et al. (2015) approach captures both first-order and second-order proximities in the graph for better embedding quality. A key limitation of these methods is their inability to leverage node features, relying solely on graph structure. Furthermore, they learn node embeddings with free parameters from the observed network in a transductive manner, meaning they cannot generalize to new nodes or networks not seen during training.

GNN-based methods leverage both network structure and node features. For the link prediction task, GNN-based approaches can be broadly divided into two categories: node-based and subgraph-based methods. In the Node-based category, models like GCN Yao et al. (2019), Graph Attention Networks (GAT) Veličković et al. (2017), GraphSAGE (SAGE) Hamilton et al. (2017), and Graph Autoencoders (GAE) Kipf & Welling (2016) represent nodes by leveraging the multi-hop structure of the graph through a message-passing mechanism. GCN utilizes convolutions operations on graphs to aggregate information from neighbors and learn node embedding. GAT assigns varying importance to neighbors using attention mechanisms when aggregating information. GAE uses an Encoder-Decoder framework to learn node embeddings, where the encoder maps nodes to embedding, and the decoder reconstructs the graph structure. GraphSAGE samples and aggregates features from a node's local neighborhood using neural networks.

Recent research studies have focused on subgraph-based methods, which integrate GNNs with enclosing subgraphs extracted from target node pairs, demonstrating remarkable effectiveness. The Weisfeiler-Lehman Neural Machine (WLNM) Zhang & Chen (2017) was among the first to apply subgraph-based GNN approaches for link predictionZhang & Chen (2018). Subgraph-based methods integrate additional information, such as subgraph features and common neighbor information to gain a deeper understanding of the relationships between nodes in predicted links. Well-known methods in this category include SEAL Zhang & Chen (2018), BUDDY Chamberlain et al. (2022), mLink Cai & Ji (2020), LGLP Cai et al. (2021), LGCL Zhang et al. (2023), DE-GNN Li et al. (2020), and NBFNet Zhu et al. (2021). The subgraph-based methods, such as SEAL Zhang & Chen (2018), extract an h-hop enclosing subgraph around the target link, learning a representation tailored to that subgraph.

## 3 PRELIMINARIES

In this section, we state the problem of link prediction and provide the formal definitions for the concepts of graphs, h-hop enclosing subgraph, line-graph, Multi-scale graph.

**Graph.** Let $G = (V, E, \mathcal{A})$ be an undirected graph, where $V = \{1, 2, \ldots, n\}$ is the set of $n$ vertices, and $E \subseteq V \times V$ is the observed edge set, which represents observed relationships and forms a subset of the complete link set $E^*$. The tensor $\mathcal{A} \in \mathbb{R}^{n \times n \times k}$ encapsulates the features of both nodes and edges. In this representation, diagonal entries $\mathcal{A}_{i,i,:}$ capture the node attributes, while off-diagonal entries $\mathcal{A}_{i,j,:}$ store the edge features. Additionally, we define $\mathbf{A} \in \{0, 1\}^{n \times n}$ as the adjacency matrix, where $\mathbf{A}_{i,j} = 1$ if and only if there exists an edge $(i, j) \in E$, and the matrix $\mathbf{X} \in \mathbb{R}^{n \times k}$, as the matrix of node features where $\mathbf{X}_i = \mathcal{A}_{i,i,:}$ for each node $i \in V$. Without node or edge attributes, we simply set $\mathcal{A} = \mathbf{A}$, treating the adjacency matrix as the feature tensor. Otherwise, the adjacency matrix $\mathbf{A}$ is derived from the first slice of $\mathcal{A}$, such that $\mathbf{A} = \mathcal{A}_{:,:,1}$.

**Link Prediction Problem.** The link prediction task is framed as designing a link predictor that operates on an observed subgraph $G \subset G^*$, defined as $\text{LP}(G) = \Pi : V \times V \to \{\text{True}, \text{False}\}$, which classifies the existence of links in the set of candidate edges $E_c$. The goal of LP is to estimate the likelihood of a potential connection between two nodes, $u$ and $v$, leveraging both the structural

characteristics of the graph and the feature information provided by $\mathcal{A}$.Mathematically, this can be expressed as $p(u, v) = p(u, v|G, \mathbf{X})$, where $\mathbf{X}$ is the node feature matrix derived from the diagonal entries of $\mathcal{A}$. While traditional methods relied on heuristic approaches to estimate $p(u, v)$, contemporary techniques employ a learnable function $f$, parameterized by $\Theta$, enabling a more flexible and data-driven estimation: $p(u, v) = f(u, v|G, \mathbf{X}, \Theta)$. These advanced methods, often implemented through GNNs, capture complex patterns in the graph structure and feature representations, improving their ability to identify potential links, specifically true missing links. The main objective is to create a vector for each edge in the graph, which captures relevant features or characteristics of the nodes and their relationships. This vector is then fed into a binary classifier that predicts the likelihood of the presence of a given edge, or in other words, predicts whether a target node pair is likely to be connected by a true missing link in the future while avoiding misclassification of false missing links. To train the classifier, two sets of edges are used: positive samples and negative samples. Positive samples are those edges that currently exist in graph G, while negative samples are a set of pairs randomly sampled from the graph where no edge currently exists.

**h-hop Enclosing Subgraph**. The h-hop enclosing subgraph for a node pair $(u, v)$ is the subgraph induced by the set of nodes within $h$-hops of either $u$ or $v$, i.e., nodes that are at most $h$-hops away from either $u$ or $v$. Specifically, the h-hop enclosing subgraph $G^h_{(u,v)}$ is represented as $G^h_{(u,v)} = (V^h_{(u,v)}, E^h_{(u,v)})$, where $V^h_{(u,v)}$ is the set of nodes within $h$-hops of $u$ or $v$, and $E^h_{(u,v)}$ is the set of edges between these nodes in the original graph. The node set $V^h_{(u,v)}$ consists of all nodes $x \in V$ that satisfy $V^h_{(u,v)} = \{x \in V : d(x, u) \leq h \text{ or } d(x, v) \leq h\}$, where $d(x, y)$ represents the shortest-path distance between nodes $x$ and $y$ in the graph $G = (V, E)$. This set includes all nodes that are reachable from either $u$ or $v$ within $h$-hops. Additionally, the edge set $E^h_{(u,v)}$ includes all edges $(x, y) \in E$ such that both $x$ and $y$ belong to $V^h_{(u,v)}$, Formally expressed as $E^h_{(u,v)} = \{(x, y) \in E : x, y \in V^h_{(u,v)}\}$. These edges represent connections between nodes within the $h$-hop neighborhood of $u$ and $v$.

**Multi-Scaled Graph**. A multi-scaled graph $SG = (V_s, E_s)$ can be defined through several key steps, starting with the original graph $G = (V, E)$, where $V$ is the set of nodes and $E$ is the set of edges. The first step involves a coarse-graining process, in which a similarity measure $S : V \times V \to \mathbb{R}$ is defined to quantify the similarity between pairs of nodes. Various similarity measures can be utilized; specifically for the link prediction task, the similarity of a group of nodes is determined by their proximity to the target nodes. Next, a partitioning function $P : V \to C$ is established to assign each node $v \in V$ to a cluster $c \in C$ based on the similarity measure, where $C$ represents a set of clusters or hyper-nodes, denoted as $C = \{C_1, C_2, \dots, C_k\}$. Nodes are then grouped into hyper-nodes according to predefined criteria, such that $\forall u, v \in V$, if $S(u, v) \geq \theta$, then $P(u) = P(v)$, where $\theta$ is a threshold for similarity. The vertex set of the scaled graph $SG$ contain the hyper-nodes, defined as $V_s = \{c_i : c_i \in C\}$. The edge set $E_s$ is constructed by connecting hyper-nodes that represent original nodes sharing edges in $G$, expressed as $E_s = \{(c_i, c_j) : \exists u \in c_i, \exists v \in c_j, (u, v) \in E\}$. Finally, the process can be recursively repeated to create multiple scales $SG_1, SG_2, SG_3$ for different levels $l$ in the hierarchy, where the $l$-th scaled graph is defined as $SG_l = (V^{(l)}_s, E^{(l)}_s)$. Each level $l$ represents a different granularity of the original graph, capturing varying patterns of interactions. Transferring a graph to a new scale reduces the complexity of the graph while preserving structural information. Fig 1 shows the process of generating graphs in different scales. The hierarchical

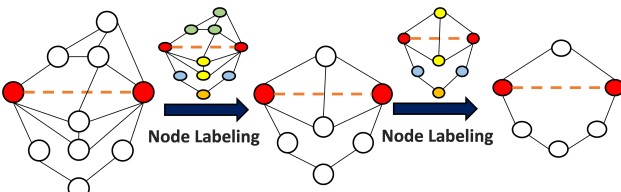

Figure 1: Example of generating graph in different scales. Similar nodes group together.

structure of a graph enables analyzing the graph at varying granularity levels. Rather than treating every individual node independently, we can group nodes based on their relations, allowing us to capture broader patterns and features that might be more insightful for classification tasks. This approach allows us to extract richer information by considering not just individual nodes, but also their collective interactions within subgraphs or clusters, leading to a more nuanced understanding of

the graph's structure and content for classification purposes. When a node's representation closely mirrors that of its local neighborhood, it struggles to gather information effectively from distant neighbors. Consequently, the surrounding subgraph may contain repetitive or unnecessary details, which can negatively impact the performance of models designed for link prediction. In simpler terms, if a node's features are too similar to those of its nearby nodes, it may miss out on important information from farther away, potentially leading to less accurate predictions in link prediction tasks.

**Line-Graph**. A line-graph $L(G)$ of a graph $G = (V, E)$ is a graph where each node in $L(G)$ corresponds to an edge in $G$, and two nodes in $L(G)$ are adjacent if and only if their corresponding edges in $G$ share a common vertex. Formally, given a graph $G = (V, E)$, the line-graph $L(G) = (V_L, E_L)$ can be described as follows: the vertex set $V_L$ consists of the edges of the original graph $G$, meaning $V_L = E$. The edge set $E_L$ is defined as $E_L = \{(e_1, e_2) \in V_L \times V_L : e_1 = (u, w), e_2 = (w, v)$ for some $u, v, w \in V\}$. This implies that two nodes $e_1 = (u, w)$ and $e_2 = (w, v)$ in $L(G)$ are connected if the edges $e_1$ and $e_2$ in $G$ share a common vertex $w$. In other words, two nodes in $L(G)$ are connected if their corresponding edges in $G$ share at least one common vertex. If the G has edges $(u, w)$ and $(w, v)$, then in the L(G), there will be nodes corresponding to these edges. Two nodes in the L(G) are connected if their corresponding edges in the G share a common node. For example, if edges $(u, w)$ and $(w, v)$ are present in the G, then in the L(G), the nodes corresponding to these edges will be connected because they share the common node $w$. Fig. 2 illustrates the line-graph derived from the original graph. This approach involves representing a link as a new node in a graph and then calculating the representation of this new node to serve as a proxy for the original link. This concept, known as line-graph transformation, treats edges (links) in the original graph as nodes in a derived line-graph, thereby capturing relationships between edges through the derived graph's node representations. In this study, inspired by a method proposed in Cai et al. (2021), we utilized the line-graph to convert the link prediction task to the node classification by using the line-graph.

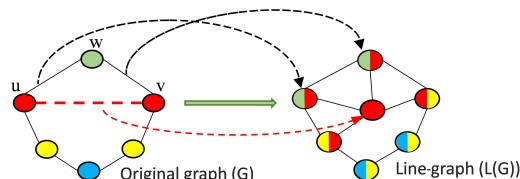

Figure 2: Demonstration of the line-graph conversion process. In the line-graph, each node corresponds to a specific edge in the original graph and is labeled with the identifiers of its two endpoints.

## 4 PROPOSED METHODOLOGY

In this section, we introduce the Multi-Scale line-graph Link Prediction (MLGLP) framework, depicted in Fig. 3. First, we group nodes with similar characteristics and connections. This consolidation allows us to transform the graph into a new scale, enabling a more efficient representation. Next, we convert the graph into a line-graph, creating three distinct line-graphs at different scales, which provide rich hierarchical structural information. Using GCN, we implement a message-passing mechanism to capture local collaborative patterns among nodes. We then focus on the embedding of target nodes within each line-graph, corresponding to target edges in each multi-scale graph derived from the original graph. By concatenating these embedding vectors, we reframe the problem from binary graph classification into a binary node classification task. To accomplish this transformation, we employ two fully connected layers as a binary classifier. Thus, we introduce a novel GNN designed to learn comprehensive relationships and features from subgraphs at different scales. This approach captures a deeper understanding of the underlying structure and dynamics of the data. The following section describes the MLGLP method in detail.

### 4.1 SUBGRAPH EXTRACTION

Detecting the presence of a link between two nodes relies on examining the topology of the graph centered around them. While leveraging global graph structural information often improves the performance, subgraph-based methods typically limit the 2-hop neighbours to balance performance

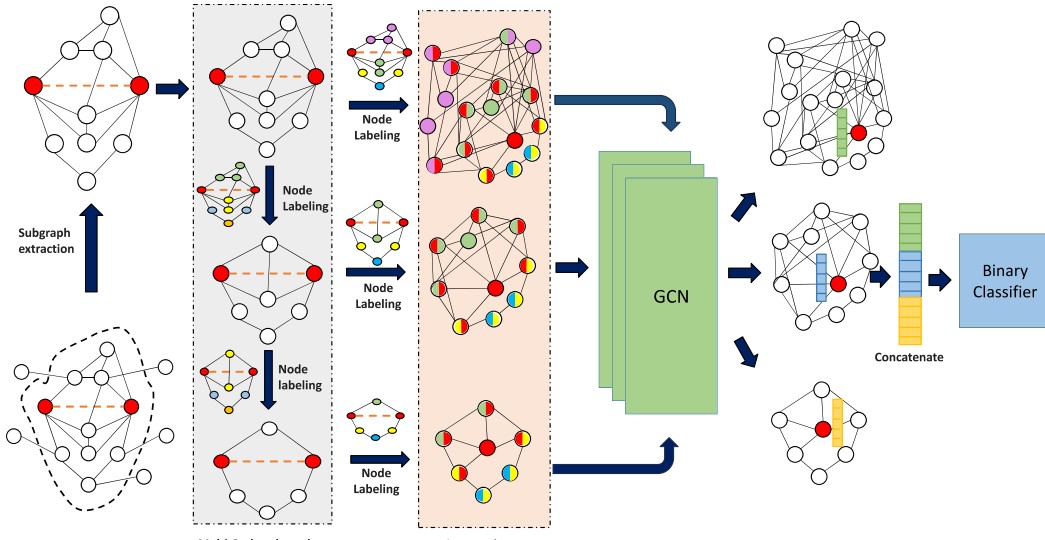

Figure 3: Overall structure of the MLGLP Framework. The process begins by extracting the enclosing subgraph from target pair nodes. Then group nodes with similar characteristics and connections, effectively merging them into single nodes. These graphs are then transformed into a line-graphs, generating three distinct line-graphs at varying scales. Using Graph Convolutional Networks (GCNs), a graph-based message-passing mechanism captures local collaborative patterns among nodes. Focus is placed on the embeddings of target nodes corresponding to target edges within each multi-scale graph derived from the original graph. These embedding vectors are concatenated, reframing the problem from binary graph classification to binary node classification. A binary classifier, implemented with two fully connected layers, is employed to learn comprehensive relationships and features of subgraphs at different scales.

and computational cost. We extract a subgraph containing the target nodes, along with all nodes connected to them within a distance of 1 or 2. For subgraph extraction, we include the target edge even for negative samples, as this step is necessary for later conversion of the subgraph into a line-graph.

## 4.2 NODE AGGREGATION - MULTI-SCALE GRAPH TRANSFORMATION

Following common GNN-based link prediction models Zhang & Chen (2018), after extracting the h-hop enclosing subgraph $G^h_{(v_i,v_j)}$ of target pair node $(v_i, v_j)$, we map $G^h_{(v_i,v_j)}$ to three different scales and form coarse-grained graphs $SG_1, SG_2, SG_3$. We aim to construct these coarse-grained scales by grouping nodes with similar connections. To do this, we measure the similarity between nodes and assign labels based on their proximity to the target nodes. This *labelling process* is crucial for predicting the presence of a link between the target nodes. Target nodes receive label of 1, while others are labeled based on their distance from the target nodes. Neighbour nodes with the same labels ($f(i) = f(j)$) are grouped together to create a hyper node in a subgraph. Equation (1) is used to assign labels to nodes.

$$f_i(u) = 1 + \min(d(u, v_i), d(u, v_j)) + d(u, v_i) + d(u, v_j) \tag{1}$$

where the variables $d(u, v_i)$ and $d(u, v_j)$ represent the shortest distances between a node $u$ and target pair nodes, $v_i$ and $v_j$, respectively. By aggregating nodes, we form a subgraph, which is then relabeled and further aggregated iteratively to produce three subgraphs representing varying detail levels.

## 4.3 LINE-GRAPH TRANSFORMATION

In this stage, we transform subgraphs $SG_1$, $SG_2$, $SG_3$ into three line-graphs $L(SG_1)$, $L(SG_2)$, $L(SG_3)$, enriching our understanding of the hierarchical structural information within the data.

Each node's label in the subgraph is derived from the computation defined by (2), and these labels are then encoded as one-hot vectors.

$$h_l(u) = 1 + \min\left(d(u, v_i), d(u, v_j)\right) + \left\lfloor \frac{d(u, v_i)}{2} \right\rfloor \left[ \left\lfloor \frac{d(u, v_i)}{2} \right\rfloor + d(u, v_j)\%2 - 1 \right] \quad (2)$$

The *labelling function* must be able to distinguish and identify two specific nodes within the subgraph. It should also be able to assign a label that reflects how important or relevant each node is in relation to the two target nodes. This involves considering the position and role of each node in the overall structure of the subgraph. Next, each subgraph is transformed into a *line-graph*, where subgraph edges become nodes in the line-graph. Then we assign a label to each node of the line-graph and use them as the initial features for the target link (target node in the line-graph). This process is applied for every edge in the original graph using graph transformation function $T(.)$ in (3).

$$T(v_i, v_j) = \text{Concat}(\min(h_l(v_i), h_l(v_j)) + \max(h_l(v_i), h_l(v_j))) \quad (3)$$

where $h(v_i)$ and $h(v_j)$, computed using (2), are used as the node representations of $v_i$ and $v_j$, respectively, after being encoded as one-hot vectors. $v_i$ and $v_j$ denote the two endpoints of an edge. The Concat$(\cdot)$ operation represents the concatenation of the two inputs, $v_i$ and $v_j$, combining their information into a single feature vector. We merge the two one-hot vectors of nodes into a single, order-invariant vector to create their feature representation. This approach allows the edge attributes to be used as node attributes in the line-graph, thereby preserving the structural information.

### 4.4 Loss Function

We apply GCN to each line-graph to generate the representation of its nodes. We focus on the embeddings of target nodes within each line-graph, particularly those corresponding to target edges for pairs of nodes within each multi-scale graph. Therefore, by concatenating these embeddings, the link prediction task is transformed into a binary node classification task.

Using node embeddings in the line-graph allows us to predict the likelihood of a potential link in the network, framing the task as a binary node classification problem. To achieve this, we employ two fully connected layers as a binary classifier, each with a dimensionality of 32.

In this paper, binary cross-entropy is used as an objective function to treat the link prediction task as a binary classification problem. The training process minimizes the cross-entropy loss across all training links. The loss function is defined as:

$$L = -\sum_{t=1}^{N} \left(y_t \log(\hat{y}_t) + (1 - y_t) \log(1 - \hat{y}_t)\right) \quad (4)$$

where $N$ represents the total number of target links used for training, $y_t$ and $\hat{y}_t$ denote the true label value and predicted probability value of the $t^{\text{th}}$ sample, respectively, indicating whether the link exists or not. The function $\log(\cdot)$ corresponds to the natural logarithm. The pseudocode of the MLGLP algorithm is provided in Appendix C.

## 5 Performance of MLGLP on benchmark datasets

In this section, we evaluated our method (MLGLP) and compared it with 12 other methods including CN Newman (2001), AA Adamic & Adar (2003), RA Zhou et al. (2009), PPR , Shortest Path Liben-Nowell & Kleinberg (2003), Katz Katz (1953), GCN Yao et al. (2019), GAE Kipf & Welling (2016), LGLP Cai et al. (2021), SEAL Zhang & Chen (2018), MLP, and MF Menon & Elkan (2011) across seven datasets. Due to space constraints, detailed descriptions of the baselines are provided in Appendix E. The results in terms of Average Precision (AP), Area Under the Curve (AUC), and loss show that MLGLP significantly outperforms other methods, demonstrating its effectiveness in link prediction tasks. detailed of the evaluation metrics are provided in Appendix D.

**Datasets.** In this study, we evaluate the MLGLP method on a diverse set of 7 datasets including Celegans, USAir, Power, NSC, Cora, Citeseer, and Router. Our experiments cover graphs of different magnitudes, encompassing variations in both node count and edge connections. Our goal is to

demonstrate the broad applicability of our method across diverse datasets of varying scales, affirming its versatility and efficacy in addressing real-world challenges. The characteristics and statistics of the datasets are presented in Table 5 of Appendix A, with further information provided in the same appendix.

## 5.1 SETTINGS

In our experiments, we set test-ratio as 0.2, which means the datasets were randomly divided in the following manner: 80% were allocated for training, and 20% were reserved for testing. All learning-based methods are trained for 50 epochs. Also, the batch number is set as 50. Additionally, experiments were conducted ten times, and the results were averaged. The damping factor is set to 0.05 for Katz, and 0.85 for PPR. The learning rate for MF and MLP is set as 0.01. For a fair comparison especially with SEAL and LGLP, we set all parameters as mentioned in the original papers. The output feature dimension for all three graph convolution layers is configured to be 32. for MLGLP the output feature dimension is set to 3*32 due to different scaled subgraphs. All experiments were conducted on AWS EC2 ml.p3.2xlarge instances equiped with 1 NVIDIA V100 GPU, 8 vCPUs, 61 GB of RAM.

## 5.2 RESULTS

The resulting average AP are presented in Table 1. It is evident that Heuristic-based methods fail to deliver satisfactory performance across all datasets due to their manually designed functions, which are unable to handle diverse cases effectively. Moreover, results show that embedding-based methods exhibit varying performance across different datasets. Additionally, since the methods are applied to plain graphs without node features, the performance of node-based GNNs decreases. Based on the results from Table 1 subgraph-based GNNs achieve the best performance. It indicates that they are capable of automatically learning the link representations from the datasets.

Table 1: Average Precision (AP) on Six Datasets for All Baseline Methods for Test-Ratio = 0.2

| Methods | Celegans | Power | Router | USAir | NSC | Cora |
|---|---|---|---|---|---|---|
| **Heuristics** | | | | | | |
| CN | 78.04% | 56.88% | 55.02% | 93.93% | 96.00% | 68.66% |
| AA | 85.45% | 57.31% | 55.16% | 94.90% | 96.81% | 69.74% |
| RA | 87.22% | 57.41% | 55.46% | 94.67% | 96.31% | 70.27% |
| PPR | 80.28% | 76.37% | 64.67% | 90.38% | 97.99% | 87.96% |
| Shortest Path | 67.35% | 74.93% | 61.42% | 75.82% | 96.05% | 84.15% |
| Katz | 87.23% | 75.36% | 63.85% | 93.86% | 98.22% | 85.40% |
| **Embedding** | | | | | | |
| MF | 84.41% | 63.00% | 82.55% | 94.81% | 99.35% | 72.95% |
| MLP | 65.38% | 51.96% | 61.69% | 83.72% | 93.21% | 57.22% |
| **Node-based GNN** | | | | | | |
| GCN | 79.00% | 59.66% | 69.35% | 92.71% | 99.07% | 69.95% |
| GAE | 70.68% | 58.16% | 55.55% | 82.05% | 72.57% | 59.92% |
| **Subgraph-based GNN** | | | | | | |
| SEAL | 86.63% | 86.71% | 97.31% | 96.44% | 99.65% | 94.80% |
| LGLP | 89.38% | 93.71% | 99.09% | 97.23% | 99.79% | 96.20% |
| MLGLP | **93.13%** | **94.96%** | **99.20%** | **98.28%** | **99.89%** | **96.23%** |

In terms of AP, our approach achieves the best performance across all datasets, with the LGLP method securing the second-best performance. Specifically, for Celegans, USAir, and Router datasets, the AP values for our method are 93.13, 98.29, and 99.20, respectively, while for the LGLP method, these values are 89.38, 97.23, and 99.09. This demonstrates that our proposed method can effectively learn superior features to represent the target link for prediction in the line-graph space.

In sub-graph-based approaches, the primary focus is on extracting the enclosing subgraph around target nodes to effectively represent them based on the structure of each subgraph. In subgraph-based GNNs, each subgraph is treated independently as a training sample. Therefore, the presence of test

edges in one subgraph does not influence other subgraphs. However, if test edges are masked, it hinders the accurate calculation of the structure of positive and negative samples, which can impact the learning process. To evaluate the impact of masking test edges, we conducted experiments analyzing the performance of sub-graph-based methods, specifically SEAL, LGLP, and MLGLP. The results in Table 2 indicate that masking the test data reduces performance and may introduce inaccuracies in learning patterns for both positive and negative samples. However, our proposed method, MLGLP, consistently demonstrates superior performance across all datasets compared to SEAL and LGLP.

Table 2: Average Precision (AP) for Masked and Unmasked Test Data with Test Ratio = 0.2

| Type | Methods | Celegans | Power | Router | USAir | NSC | Cora |
|------|---------|----------|-------|--------|-------|-----|------|
|        | SEAL  | 83.12% | 77.73% | 91.10% | 95.33% | 99.61% | 89.03% |
| Masked | LGLP  | 88.25% | 84.66% | 93.43% | **96.21%** | 99.65% | 93.12% |
|        | MLGLP | **90.15%** | **87.01%** | **94.25%** | 96.20% | **99.78%** | **93.60%** |
|          | SEAL  | 86.63% | 86.71% | 97.31% | 96.44% | 99.65% | 94.80% |
| Unmasked | LGLP  | 89.38% | 93.71% | 99.09% | 97.23% | 99.79% | 96.20% |
|          | MLGLP | **93.13%** | **94.97%** | **99.18%** | **98.29%** | **99.89%** | **96.23%** |

Fig. 4(a) illustrates the training loss over 50 epochs. It is evident that MLGLP outperforms both LGLP and SEAL and achieves lower loss compared to other methods. This suggests that our proposed approach learns more effective features for representing target links in the line-graph space. Specifically, MLGLP gathers more information from different scales during the training process, enabling it to extract complex features crucial for accurate predictions. In contrast, LGLP performs better than SEAL but does not reduce the loss as effectively as MLGLP over the training period. LGLP converges quickly but struggles to extract complex features effectively during the learning phase, as depicted in the figure. This highlights the superior capability of MLGLP in leveraging training data to enhance feature representation for link prediction tasks. Fig. 4(b) shows the AUC comparison between LGLP, SEAL, and MLGLP methods for Celegans dataset. The results clearly demonstrate that our proposed model significantly outperforms SEAL and LGLP in terms of achieving a higher AUC.

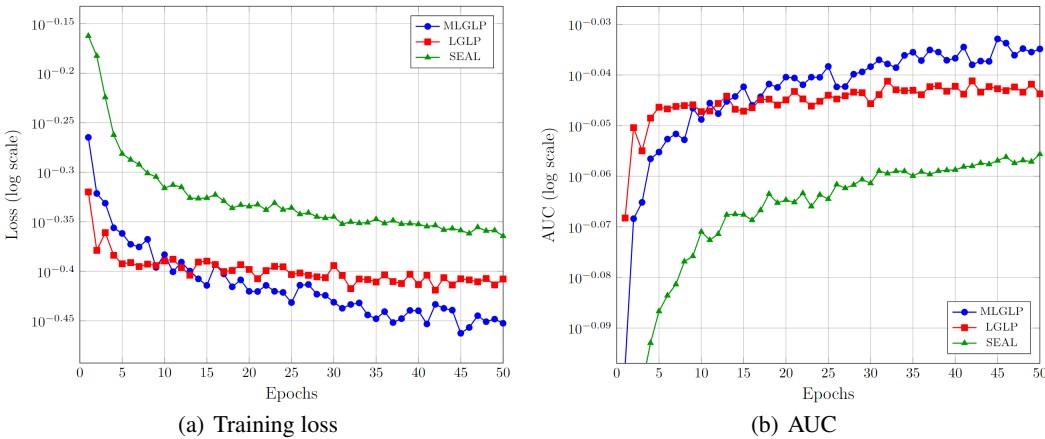

(a) Training loss      (b) AUC

Figure 4: Comparison of training loss and AUC between the LGLP, SEAL, and MLGLP methods on the Celegans dataset.

To highlight the performance of our proposed method, we extracted edge features from the penultimate fully connected layer and applied t-distributed stochastic neighbor embedding (t-SNE) for visualization. Fig. 5 illustrates the results on the Router, Cora, and Citeseer datasets, focusing on a 0.2 test ratio. Positive links are depicted in Red and negative links in Blue. The visualization clearly demonstrates that the features learned by our model form well-separated clusters, making the classification of positive and negative links remarkably straightforward. This showcases the strong discriminative power of our approach.

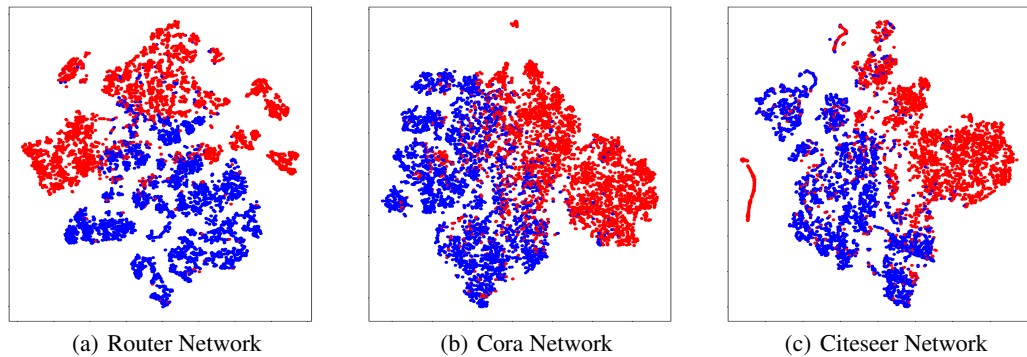

| (a) Router Network | (b) Cora Network | (c) Citeseer Network |

Figure 5: t-SNE visualization of the Router, Cora, and Citeseer datasets for the MLGLP method. Red indicates positive links, while blue represents negative links.

## 6 ABLATION STUDY

Table 3 presents the Average Precision (AP) and Area Under the Curve (AUC) scores for our proposed MLGLP framework across six datasets, evaluated using different scales and a combination of all scales. The test ratio for this evaluation was set to 0.2, and we aimed to examine the contribution of each scale within the multi-scale approach. Evaluating individual scales indicates that each scale contributes valuable information, highlighting the significance of capturing different levels of structural patterns within the graph. Depending on the dataset, individual scales can sometimes yield competitive results, particularly on the NSC and Router datasets. When all scales are used together (the "All" method), the model achieves the highest performance on most datasets. For instance, on the USAir dataset, we observe an AP of 98.28% and an AUC of 98.31%. Also the Celegans dataset achieves AP and AUC values of 93.13% and 90.76%, respectively. These results demonstrate that leveraging all scales together consistently provides the best results, confirming the robustness of the multi-scale approach.

Table 3: Average Precision (AP) and AUC on Six Datasets for MLGLP with Different Scales (Test Ratio = 0.2): Evaluating the Impact of Each Scale in a Multi-Scale Approach.

| Methods | Celegans | Power | Router | USAir | NSC | Cora | Citeseer |
|---------|----------|-------|--------|-------|-----|------|----------|
| **AP** | | | | | | | |
| All | **93.13%** | **94.96%** | **99.20%** | **98.28%** | **99.89%** | **96.23%** | 96.25% |
| Scale-1 | 89.38% | 93.71% | 99.09% | 97.23% | 99.79% | 96.20% | 95.86% |
| Scale-2 | 88.65% | 89.14% | 97.07% | 95.87% | 99.32% | 93.43% | 93.43% |
| Scale-3 | 75.67% | 89.53% | 97.06% | 93.32% | 99.33% | 93.50% | 93.50% |
| **AUC** | | | | | | | |
| All | **90.76%** | **93.84%** | **99.11%** | **98.31%** | 99.68% | **95.79%** | 95.43% |
| Scale-1 | 90.75% | 92.10% | 99.05% | 98.14% | **99.82%** | 95.24% | 94.47% |
| Scale-2 | 88.11% | 87.93% | 97.29% | 95.75% | 99.61% | 92.35% | 92.34% |
| Scale-3 | 75.33% | 87.34% | 97.27% | 92.81% | 99.37% | 92.42% | 92.42% |

## 7 CONCLUSION AND FUTURE RESEARCH

In this study, we explored learning-based methods for link prediction. Specifically, we propose a novel approach using GNNs called Multi-Scale line-graph Link Prediction (MLGLP). This method aims to effectively learn the graph structure and extract representative features from edges, addressing challenges such as information loss and handling multi-scale information. To facilitate hierarchical learning, our approach involves constructing coarse-grained graphs at three distinct scales, thereby revealing complex relationships within the data. Furthermore, to accommodate GNN learning across multi-scale graphs with varying node and edge sizes, we transform the graphs into line-graph representations. This transformation allows us to learn node embeddings within each subgraph

and translates the link prediction task into a node classification problem. Experimental results indicate promising performance enhancements compared to heuristics, embeddings, node-based GNNs, and sub-graph-based GNNs link prediction methods, especially SEAL, and LGLP. A possible future research is expanding this methodology to heterogeneous graphs and using the valuable heterogeneity information.

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

## A  NOTATIONS

This section provides an overview of the symbols and notations utilized in this paper, with a detailed summary provided in Table 4.

**True Missing Links** are edges that should exist in the graph but are not currently observed. These edges are part of the complete graph's true edge set $E^*$ but are not included in the observed edge set $E$. Formally, a true missing link $(u, v)$ satisfies:

$$\{(u, v) \in E^* \text{ and } (u, v) \notin E\}$$

where $E$ is the set of obsserved edges and $E^*$ is the set of all true edges in the complete graph. For example if $E^*$ includes an edge $(u, v)$ but $(u, v)$ is not in $E$, then $(u, v)$ is a true missing link.

**False Missing Links** are pairs of nodes that are incorrectly considered as potential links but are not part of the true edge set $E^*$. These are often pairs where a link is not present in both the complete graph and the observed graph. Formally, a false missing link $(u, v)$ satisfies:

$$\{(u, v) \mid u, v \in V \text{ and } (u, v) \notin E^* \text{ and } (u, v) \notin E\}$$

for example if $E^*$ does not include an edge $(u, v)$ and $(u, v)$ is also not in $E$, this edge can be mistakenly considered a potential link (false positive) by some models.

**Positive Samples** are defined as edges that exist in both the observed edge set $E$ and the complete graph $E^*$. In other words, these are edges that are part of the true edge set and are observed. Formally, these are edges in $E \cap E^*$.

**Negative Samples** are defined as edges that do not exist in either the set of true edges $E^*$ or the set of observed edges $E$. They can be considered as false positives, i.e., pairs of nodes that should not be connected. These correspond to pairs of nodes that are not connected by an edge in the graph. Since identifying true negative samples—edges that should never exist—is often challenging due to the lack of complete knowledge about the graph or underlying data distribution, a common strategy is randomly selecting negative samples from the set of nonexistent edges. Thus, the negative samples are selected randomly from edges that are not in $E$, acknowledging that while these may not be

Table 4: Summary of Notations Used in the Paper

| Notations | Definitions or Descriptions |
|---|---|
| $G = (V, E, X)$ | Graph with node set $V$, edge set $E$, and node features $X$ |
| $n$ | Number of nodes, $n = |V|$ |
| $m$ | Number of edges, $m = |E|$ |
| $E^*$ | Complete set of possible edges between nodes in $V$ |
| $\mathcal{A}$ | Feature tensor, $\mathcal{A} \in \mathbb{R}^{n \times n \times k}$, capturing both node and edge features |
| $\mathcal{A}_{i,i,:}$ | Node features for node $i$ |
| $\mathcal{A}_{i,j,:}$ | Edge features for edge $(i, j)$ |
| $\mathbf{A}$ | Adjacency matrix, $\mathbf{A} \in \{0, 1\}^{n \times n}$ |
| $\mathbf{A}_{i,j}$ | Adjacency matrix entry, 1 if edge $(i, j)$ exists, otherwise 0 |
| $\mathbf{X}$ | Node feature matrix, $\mathbf{X} \in \mathbb{R}^{n \times k}$ |
| $\mathbf{X}_i$ | Feature vector for node $i$, $\mathbf{X}_i = \mathcal{A}_{i,i,:}$ |
| $G^*$ | Complete graph with all possible edges $E^*$ between nodes in $V$ |
| $E_c$ | Set of candidate edges for link prediction |
| $p(u, v)$ | Probability of a link between nodes $u$ and $v$ |
| $f(u, v|G, \mathbf{X}, \Theta)$ | Learnable function to estimate $p(u, v)$, parameterized by $\Theta$ |
| $\Theta$ | Parameters of the learnable function $f$ |
| $h$ | Maximum number of hops in the h-hop enclosing subgraph |
| $G^h_{(u,v)}$ | h-hop enclosing subgraph for node pair $(u, v)$ |
| $V^h_{(u,v)}$ | Node set within $h$-hops of either $u$ or $v$ |
| $E^h_{(u,v)}$ | Edge set within $h$-hops of either $u$ or $v$ |
| $d(x, y)$ | Shortest-path distance between nodes $x$ and $y$ |
| $SG = (V_s, E_s)$ | Multi-scaled graph with vertex set $V_s$ and edge set $E_s$ |
| $S(u, v)$ | Similarity measure between nodes $u$ and $v$ |
| $P(v)$ | Partitioning function assigning node $v$ to a cluster |
| $C$ | Set of clusters or hyper-nodes |
| $\theta$ | Threshold for node similarity |
| $SG_l = (V_s^{(l)}, E_s^{(l)})$ | Scaled graph at level $l$ in a hierarchical structure |
| $L(G) = (V_L, E_L)$ | Line-graph of graph $G$ |
| $V_L$ | Node set of the line-graph |
| $E_L$ | Edge set of the line-graph |
| $L$ | Loss function |
| $N$ | Total number of target links used for training |
| $y_t$ | True label value for the $t^{\text{th}}$ sample |
| $\hat{y}_t$ | Predicted probability value for the $t^{\text{th}}$ sample |
| $f_i(u)$ | Label of node $u$ |
| $h(u)$ | Node representation of node $u$ |
| $T(v_i, v_j)$ | Function for concatenating features of nodes $v_i$ and $v_j$ |

guaranteed to be true negatives, they serve as a practical and sufficient set of negative examples for training the model.

**Candidate Set** ($E_c$) is constructed to include both types of samples to train and evaluate the link prediction model effectively. The goal is to differentiate between these two types of samples and correctly classify which candidate edges should be present in the graph. These candidates help train and test the model to accurately predict the presence of true missing links while avoiding false positives.

## B  DATASETS

A brief overview of the benchmark datasets utilized in this study is as follows. These datasets are used for evaluating models in various types of graphs, including social networks, citation networks, and biological networks.

- Router: This dataset represents a router-level Internet graph with 5022 nodes and 6258 edges, modeling connections between routers in the network. edges.

- Cora: This citation graph includes 2708 scientific publications and 5278 links, with a dictionary of 1433 unique words derived from the papers.

- Citeseer: This dataset features 3312 scientific publications and 4552 links, accompanied by a dictionary of 3703 unique words from the publication texts.

- USAir: This dataset represents a graph of US airlines, containing 332 nodes and 2126 edges.

- NSC: This dataset illustrates the collaboration relationships of network science researchers, containing 1589 nodes and 2742 edges.

- Celegans: This dataset contains the biological neural network of *C. elegans*, consisting of 297 nodes and 2148 edges.

- Power: This dataset illustrates the topology of the Western States Power Grid of the United States, containing 4941 nodes and 6594 edges.

Table 5: Summary Statistics of the Datasets Used in the Study

| Statistic | Router | Cora | Citeseer | USAir | NSC | Celegans | Power |
|-----------|--------|------|----------|-------|------|----------|-------|
| #Nodes | 5022 | 2708 | 3312 | 332 | 1461 | 297 | 4941 |
| #Edges | 6258 | 5429 | 4552 | 2126 | 2742 | 2148 | 6594 |
| #Features | NA | 1432 | 3703 | NA | NA | NA | NA |

## C  ALGORITHM

In this section, we present the pseudo-code of the MLGLP framework to enhance clarity and understanding.

---

**Algorithm 1:** MLGLP Algorithm

---

   **Data:** Target link $(v_i, v_j)$, graph $G$
   **Result:** Predicting the existence or nonexistence of the target link
   **Input:** $h = 2$
**1** Extract $h$-hop enclosing subgraph of target node pair $G^h_{(v_i,v_j)}$;
**2** Compute node labeling by equation (3);
**3** Transfer $G^h_{(v_i,v_j)}$ to three multi-scale subgraphs $SG_1, SG_2, SG_3$;
**4** Convert multi-scale subgraphs $SG_1, SG_2, SG_3$ to line graphs $LSG_1, LSG_2, LSG_3$;
**5** Initialize node embedding;
**6** Compute the node embedding using GCN;
**7** Concatenate embedding vectors of target nodes in $LSG_1, LSG_2, LSG_3$;
**8** Predict existence or nonexistence of the link $(v_i, v_j)$ using a binary classifier;

---

## D  EVALUATION METRICS

This section provides detailed descriptions of evaluation metrics including AUC, AP.

**a) AUC:** The AUC is computed as the number of successful predictions divided by the total number of comparisons. Successful predictions can be determined based on scores for each node pair using predefined heuristics (e.g., common neighbors) or probabilities for each node pair using the GNN model. Compute the AUC using the formula:

$$AUC = \frac{n' + 0.5 \times n''}{n} \qquad (5)$$

Where $n$ is the total number of predictions, $n'$ is the number of successful predictions (i.e., the number of times for each positive-negative pair the positive sample has a higher score or probability than the negative sample), and $n''$ is the number of times that scores are the same.

**b) AP:** Average Precision measures the precision of a model at various threshold levels, capturing the ability to rank positive links higher than negative ones. It is formulated as:

$$AP = \sum_{k=1}^{n} P(k) \cdot \Delta r(k) \tag{6}$$

Where $n$ is the total number of positive and negative links, $P(k)$ is the precision at rank $k$, and $\Delta r(k)$ is the change in recall at rank $k$.

## E    BASELINE METHODS

This section provides detailed descriptions of the baseline methods utilized in this paper. We compare our proposed MLGLP with 12 methods spanning various categories: heuristics, embeddings, node-based GNNs, and sub-graph-based GNNs. In our study, we evaluated our proposed method against heuristic approaches such as CN, AA, PPR, Shortest Path, and, Katz, embedding techniques like MLP and MF, node-based GNN methods including GAE, GCN, and finally subgraph-based GNN like SEAL and LGLP. Table 6 shows Details of heuristic-based methods utilized in this paper.

Table 6: Details of heuristic-based methods for link prediction utilized in this paper

| Name | Formula | Order |
|------|---------|-------|
| Common Neighbors (CN) | $\|\Gamma(x) \cap \Gamma(y)\|$ | first |
| Adamic-Adar(AA) | $\sum_{z \in \Gamma(x) \cap \Gamma(y)} \frac{1}{\log \|\Gamma(z)\|}$ | second |
| Resource Allocation (RA) | $\sum_{z \in \Gamma(x) \cap \Gamma(y)} \frac{1}{\|\Gamma(z)\|}$ | second |
| PageRank(PPR) | $[\pi_x]_y + [\pi_y]_x$ | high |
| Shortest Path | $\frac{1}{length(shortestpath(x,y)}$ | high |
| Katz | $\sum_{l=1}^{\infty} \beta^l \|\text{walks}^{\langle l \rangle}(x,y)\|$ | high |

## F    VISUALIZATION

Due to space constraints, the t-SNE visualizations for the Power, USAir, and NSC datasets are also presented in Fig.6 within this section. The visualization effectively demonstrates that the features generated by our model form distinct clusters, which simplifies the differentiation between positive and negative links. This underscores the robust discriminative ability of our approach.

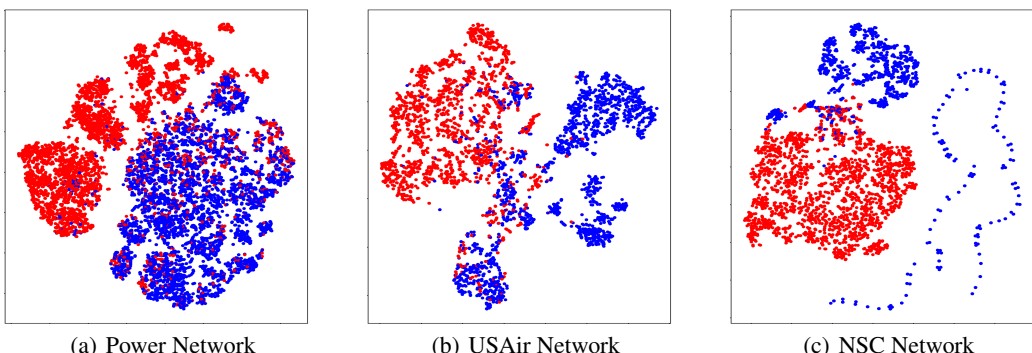

| (a) Power Network | (b) USAir Network | (c) NSC Network |

Figure 6: t-SNE visualization of the Power, USAir, and NSC datasets for the MLGLP method. Red indicates positive links, while blue represents negative links.

Also, the t-SNE visualizations for the LGLP method on all six data sets Routers, Cora, Citeseer, Power, USAir, and NSC presented in Fig.7.

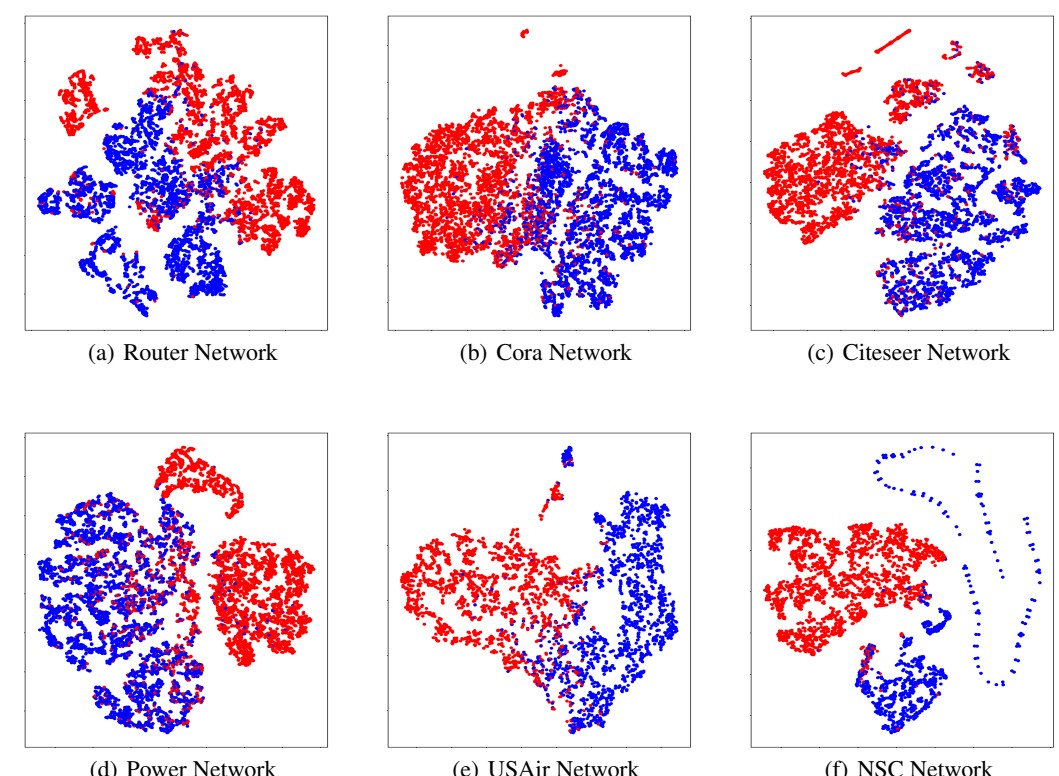

Figure 7: t-SNE visualization of the Router, Cora, Citeseer, Power, USAir, NSC datasets for the LGLP method. Red indicates positive links, while blue represents negative links.

## G AUC-BASED PERFORMANCE

The resulting average AUC are presented in Table 7. The data in the table indicates that subgraph-based GNNs excel compared to alternative approaches, showcasing their effectiveness in automatically capturing link representations from the datasets. For AUC, our method gains the best performance except for NSC which gains 99.68.

Table 7: AUCs on Six Datasets for All Baseline Methods for Test-Ratio = 0.2

| Methods | Celegans | Power | Router | USAir | NSC | Cora |
|---|---|---|---|---|---|---|
| **Heuristics** | | | | | | |
| CN | 79.93% | 56.91% | 55.05% | 94.42% | 96.11% | 68.75% |
| AA | 83.92% | 57.29% | 55.09% | 94.98% | 96.78% | 69.64% |
| RA | 86.76% | 57.40% | 55.33% | 93.99% | 96.28% | 70.33% |
| PPR | 81.41% | 62.57% | 45.83% | 89.70% | 97.99% | 82.03% |
| Shortest Path | 74.69% | 62.52% | 40.74% | 82.48% | 96.72% | 79.25% |
| Katz | 87.74% | 71.17% | 43.89% | 92.10% | 98.18% | 79.45% |
| **Embedding-based** | | | | | | |
| MF | 85.34% | 62.77% | 79.29% | 94.64% | 99.27% | 70.88% |
| MLP | 64.80% | 52.20% | 61.66% | 83.42% | 93.09% | 56.27% |
| **Node-based GNN** | | | | | | |
| GCN | 79.45% | 58.89% | 68.74% | 92.02% | 98.93% | 67.97% |
| GAE | 70.53% | 55.53% | 53.05% | 82.51% | 82.42% | 59.39% |
| **Subgraph-based GNN** | | | | | | |
| SEAL | 89.16% | 83.67% | 97.36% | 96.58% | 99.66% | 93.40% |
| LGLP | 90.75% | 92.11% | 99.05% | 98.14% | **99.82%** | 95.25% |
| MLGLP | **90.76%** | **93.84%** | **99.11%** | **98.31%** | 99.68% | **95.79%** |

# H ROBUSTNESS EVALUATION WITH REDUCED TRAINING DATA

To demonstrate the robustness of our proposed approach with reduced training data, we conducted experiments on all datasets using only 50 percent of the training links. The remaining links were used as test data. The outcomes, including AUC and AP values, are presented in Tables 8 and 9 The results consistently show that our method outperforms all baseline methods in terms of AP across all datasets and AUC in the majority of datasets. Remarkably, our method maintains strong performance even with only 50 percent of the training links, achieving AUC and AP values comparable to those obtained with 80 percent of the training links. For example, AP for Celegans, Power, Router, USAir, NSC, and Cora just reduced by 3.45%, 1.69%, 0.58%, 2.67%, 0.06%, and 0.90%, respectively.

Table 8: AUCs on Six Datasets for All Baseline Methods Using Test-Ratio = 0.5

| Methods | Celegans | Power | Router | USAir | NSC | Cora |
|---|---|---|---|---|---|---|
| Heuristics | | | | | | |
| CN | 70.87% | 53.14% | 53.39% | 87.82% | 90.38% | 59.69% |
| AA | 72.52% | 53.05% | 52.57% | 88.11% | 92.41% | 59.79% |
| RA | 72.78% | 53.28% | 52.86% | 87.77% | 92.60% | 59.12% |
| PPR | 79.51% | 57.73% | 54.32% | 85.42% | 95.56% | 68.39% |
| Shortest Path | 72.11% | 57.45% | 54.44% | 82.66% | 94.86% | 67.10% |
| Katz | 80.09% | 58.28% | 54.41% | 89.71% | 96.17% | 68.95% |
| Embedding | | | | | | |
| MF | 71.99% | 54.70% | 77.47% | 90.87% | 98.64% | 63.96% |
| MLP | 62.81% | 51.19% | 60.32% | 80.32% | 89.31% | 55.88% |
| Node-based GNN | | | | | | |
| GCN | 73.80% | 52.44% | 59.49% | 90.08% | 98.09% | 60.14% |
| GAE | 62.20% | 53.88% | 49.65% | 70.70% | 76.57% | 55.14% |
| Subgraph-based GNN | | | | | | |
| SEAL | 88.19% | 82.21% | 97.12% | 96.32% | 99.64% | 93.42% |
| LGLP | 90.94% | 91.78% | 98.98% | 97.34% | 99.77% | 95.22% |
| MLGLP | **91.52%** | **93.28%** | **98.62%** | **97.22%** | **99.83%** | **95.33%** |

Table 9: Average Precision (AP) on Six Datasets for All Baseline Methods Using Test-Ratio = 0.5

| Methods | Celegans | Power | Router | USAir | NSC | Cora |
|---|---|---|---|---|---|---|
| Heuristics | | | | | | |
| CN | 68.23% | 53.12% | 53.38% | 87.47% | 90.34% | 59.57% |
| AA | 70.72% | 53.05% | 52.55% | 88.46% | 92.44% | 59.91% |
| RA | 72.24% | 53.27% | 52.80% | 88.51% | 92.61% | 59.18% |
| PPR | 78.25% | 57.68% | 60.89% | 84.90% | 95.58% | 75.41% |
| Shortest Path | 66.00% | 57.65% | 60.64% | 77.60% | 94.47% | 73.46% |
| Katz | 79.90% | 58.42% | 61.07% | 92.03% | 96.25% | 75.71% |
| Embedding | | | | | | |
| MF | 82.25% | 54.48% | 81.38% | 91.19% | 98.86% | 66.42% |
| MLP | 64.28% | 51.19% | 60.06% | 81.35% | 89.67% | 56.84% |
| Node-based GNN | | | | | | |
| GCN | 73.06% | 53.08% | 62.04% | 90.56% | 98.33% | 62.96% |
| GAE | 61.54% | 53.96% | 49.80% | 68.44% | 66.95% | 54.55% |
| Subgraph-based GNN | | | | | | |
| SEAL | 86.58% | 85.57% | 97.06% | 95.96% | 99.51% | 94.73% |
| LGLP | 89.63% | 93.87% | 98.86% | 98.28% | 99.77% | 95.98% |
| MLGLP | **93.43%** | **95.04%** | **99.06%** | **98.54%** | **99.92%** | **96.42%** |

# I  TIME COMPLEXITY ANALYSIS

For extracting each graph $G^h_{(v_i, v_j)}$ with $n$ nodes and $m$ edges where $(v_i, v_j)$ is the target pair of nodes, total time complexity for calculating distances for both target nodes is $\mathcal{O}(n + m)$ per scale. Constructing a *line-graph* has a time complexity of $\mathcal{O}(m^2)$, where $m$ is the number of edges in the original graph (since each edge in the original graph becomes a node in the *line-graph*). Time Complexity for GCN operation depends on the number of nodes and edges in the line-graph. For a graph with $|V|$ nodes and $|E|$ edges, the complexity is $\mathcal{O}(|V| + |E|)$. Since we are dealing with line-graphs, this translates to $\mathcal{O}(n^2 + n^2) = \mathcal{O}(n^2)$ for each scale. For three scales, it's $\mathcal{O}(3n^2) = \mathcal{O}(n^2)$.

