# OpenReview forum: "MLGLP: Multi-Scale Line-Graph Link Prediction based on Graph Neural Networks"
_ICLR.cc/2025/Conference — ICLR 2025 Conference Withdrawn Submission_

### Official Review · Reviewer_LUFZ · 2024-10-28

**Soundness:** 2
**Presentation:** 2
**Contribution:** 2
**Rating:** 3
**Confidence:** 5

**Summary:**

This paper proposed a link prediction method named Multi-Scale line-graph Link Prediction (MLGLP). MLGLP used three scales to capture information at a different level of granularity. The link prediction problem is defined as a node classification problem on a line graph, which facilitates a deeper understanding of relationships within the graph. Experiments conducted on several benchmark datasets validated the effectiveness of MLGLP.

**Strengths:**

1. The authors use multi-scale subgraphs for link prediction to capture graph information at different granularities. The approach is interesting.
2. The authors transform the link prediction problem into a node classification problem on a line graph, which better addresses the issue of link representation.
3. The proposed method outperforms existing methods in the experiments.

**Weaknesses:**

1. This paper appears to be an unfinished draft, containing many textual errors (e.g., the beginning of line 349 lacks capitalization, and line 372 is missing a period) and missing sentences (e.g., line 510).
2. The baselines chosen in this paper are outdated.
3. The paper lacks novelty, as the proposed module is superficial and easy to conceive. The proposed method for converting to line graph is very similar to LGLP.

**Questions:**

1. There are many papers on link prediction in line graphs; can you explain what distinguishes MLGLP from them?
2. It's necessary to use the latest methods as baselines, such as BUDDY[1], NCNC[2], and PEG[3].
3. I have doubts about the visualization results; in Figure 5, the blue and red points seem to overlap, indicating that MLGLP cannot distinguish between positive and negative samples. An explanation is needed.

Reference:
[1] Chamberlain BP, Shirobokov S, Rossi E, Frasca F, Markovich T, Hammerla N, Bronstein MM, Hansmire M. Graph neural networks for link prediction with subgraph sketching. arXiv preprint arXiv:2209.15486. 2022 Sep 30.
[2] Wang X, Yang H, Zhang M. Neural common neighbor with completion for link prediction. arXiv preprint arXiv:2302.00890. 2023 Feb 2.
[3] Wang H, Yin H, Zhang M, Li P. Equivariant and stable positional encoding for more powerful graph neural networks. arXiv preprint arXiv:2203.00199. 2022 Mar 1.

---

> ### Author Response · Authors · 2024-11-28
>
> We are truly grateful for your insightful feedback, which has played a crucial role in enhancing and refining our work. In the following, we provide comprehensive responses to address the concerns and questions you raised under Weaknesses (W) and Questions (Q).
>
> $\textbf{(W1):}$ Thank you for your helpful feedback. To address this comment, we have carefully reviewed the paper and have corrected all the textual errors.
>
> $\textbf{(W2):}$ Thank you for your feedback on the baselines. In the revised version, we will update the baselines to include more recent and relevant methods to better reflect the current state of the field.
>
> $\textbf{(W3):}$ Thank you for your feedback. While the concept of converting to a line graph may appear similar to LGLP, our method, unlike LGLP, which focuses on a single scale, incorporates a multi-scale framework. This enables it to capture richer and more diverse structural information across various scales of the graph. Additionally, as highlighted in the paper, our approach outperforms existing methods like LGLP in terms of accuracy.
>
> $\textbf{(Q1):}$ Thank you for highlighting the numerous works on link prediction in line graphs. While existing methods have contributed significantly to the field, our proposed MLGLP (Multi-Scale Line Graph Link Prediction) framework offers key distinctions:
>
> It is a $\textbf{Multi-Scale Framework}$, Unlike traditional methods that focus on a single level of representation, MLGLP incorporates multiple scales to capture both local and global structural patterns in the line graph. This enables a more comprehensive understanding of the graph's topology, resulting in improved link prediction performance across diverse datasets.
>
> MLGLP leverages rich structural patterns from different scales to maintain robust performance. As shown in our ablation studies, each scale contributes uniquely to the model's robustness, outperforming single-scale methods in capturing nuanced link dependencies.
>
> $\textbf{(Q2):}$ Thank you for the suggestion regarding the inclusion of recent baselines like BUDDY, NCNC, and PEG.
> Thank you for your valuable feedback. In the revised version of the paper, we will certainly include a comparison of MLGLP alongside other methods such as PEG, BUDDY, and NCNC. We are confident that these additions will help validate the efficacy of MLGLP, and we will ensure these clarifications are incorporated in the revision.
> It is important to highlight that, as mentioned in Table 1 of the paper $\textbf{[1]}$, for $\textbf{Cora}$, the best  $\textbf{AUC}$ for $\textbf{PEG}$ without using node features is $\textbf{90.78 ± 0.09}$, while for $\textbf{MLGLP}$, the AUC is $\textbf{95.79}$, as highlighted in $\textbf{Table 7}$ of the appendix.  These results demonstrate the significant improvement of MLGLP over existing methods, including PEG, in terms of performance.
>
> We chose to compare MLGLP with LGLP as a baseline for the following reasons, despite other methods being noteworthy in the field.
>
> 1. LGLP is relevant to our method as it was a leading approach in GNN-based link prediction when our work began. Both LGLP and MLGLP focus on direct edge representations using localized subgraphs, making them conceptually similar. In contrast, methods like PEG, Buddy, and NCNC rely on node embeddings for edge prediction, making them less directly comparable to MLGLP.
>
> 2. Subgraph-based methods like LGLP, MLGLP, and SEAL have the advantage of localized computation, making them more efficient during inference compared to whole-graph approaches like PEG [2]. Including methods with similar computational paradigms ensures a fair evaluation.
>
> 3.LGLP, MLGLP, and SEAL explicitly generate edge-specific embeddings, which are better suited for link prediction tasks. In contrast, PEG, Buddy, and NCNC produce node embeddings and infer edge relationships indirectly, which might limit their performance for certain edge-centric tasks.
>
> $\textbf{(Q3):}$ Thank you for your insightful question. The overlap in the t-SNE visualizations (Figure 5) does not indicate poor performance in link prediction. t-SNE, a dimensionality reduction technique, may not always preserve local structures, leading to some overlap even when the model performs well. To address this, we added t-SNE results for LGLP and SEAL in the appendix. Further evidence of MLGLP's effectiveness is shown in the line graph link prediction task, where our method outperforms the baselines in classification accuracy and AUC, despite the overlap in the t-SNE plot. This demonstrates that MLGLP effectively distinguishes between positive and negative links in a higher-dimensional space.
>
> REFERENCES:
> [1] Wang H, Yin H, Zhang M, Li P. Equivariant and stable positional encoding for more powerful graph neural networks. arXiv preprint arXiv:2203.00199. 2022.
> [2] Zhang, S., Liu, Y., Sun, Y., & Shah, N. Graph-less Neural Networks: Teaching Old MLPs New Tricks Via Distillation. In Proceedings of the ICLR, 2022.

---

> > ### Comment · Reviewer_LUFZ · 2024-11-29
> > **Thank you**
> >
> > Thank you for your reply. Since you still haven't solved my doubt (Q2), I will keep my score.

---

> ### Author Response · Authors · 2024-12-03
>
> We have conducted additional experiments comparing NCNC[1] with our method (MLGLP), and the results demonstrate the superiority of our method compared to NCNC. We found that NCNC is highly sensitive to node features, which affects its performance in scenarios with limited or absent node attributes. The results are as follows:
>
>
> $\textbf{Comparison Results}$
>
> $\textbf{Cora}$
> - $NCNC$
>   - $Node Feature$: AUC = 95.72\%, AP = 95.89\%
>   - Random Node Feature: $\textbf{AUC = 70.78 \\% }$, $\textbf{AP = 75.16 \\% }$
>   - Onehot-degree-node - Node Feature: AUC = 85.06\%, AP = 88.34\%
> - $MLGLP$: $\textbf{AUC = 95.79\\%}$, $\textbf{AP = 96.23\\%}$
>
> $\textbf{NSC}$
> - $NCNC$
>   - Random Node Feature: AUC = 59.76 \%, AP = 56.87\%
>   - Onehot-degree-node - Node Feature: $\textbf{AUC = 95.39\\%}$, $\textbf{AP = 96.95\\%}$
> - $MLGLP$: $\textbf{AUC = 99.68\\%}$, $\textbf{AP = 99.89\\%}$
>
> $\textbf{USAir}$
> - $NCNC$
>   - Random Node Feature: AUC = 56.30\%, AP = 53.57\%
>   - Onehot-degree-node - Node Feature: $\textbf{AUC = 96.88\\%}$, $\textbf{AP = 96.24\\%}$
> - $MLGLP$: $\textbf{AUC = 98.31\\%}$, $\textbf{AP = 98.28\\%}$
>
> $\textbf{Router}$
> - $NCNC$
>   - Random Node Feature: AUC = 72.26\%, AP = 68.78\%
>   - Onehot-degree-node - Node Feature: $\textbf{AUC = 96.07\\%}$, $\textbf{AP = 96.26\\%}$
> - $MLGLP$: $\textbf{AUC = 99.11\\%}$, $\textbf{AP = 99.20\\%}$
>
>
> $\textbf{Advantages of MLGLP}$
>  - $\textbf{Independence from Node Features}$
>  - $\textbf{Inference time Efficiency}$
> - $\textbf{Task-Specific Design}$
>
> As mentioned in the NCNC paper[1], NCNC outperforms PEG and BUDDY. Therefore, I have compared our method to NCNC. We will include detailed comparisons in the camera-ready version to further illustrate these distinctions and validate our method against NCNC, PEG, and BUDDY.
>
> We hope this clarification addresses your concerns and would greatly appreciate it if you could reevaluate our contribution in light of these new results.
> Thank you for your time and consideration.
>
>
> Reference: [1] Wang X, Yang H, Zhang M. Neural common neighbor with completion for link prediction. arXiv preprint arXiv:2302.00890. 2023 Feb 2.

---

> ### Comment · Reviewer_LUFZ · 2024-12-03
> **Reply to Authors**
>
> Thank you for your reply. I still have four concerns:
> 1. In the case of datasets with existing features, such as Cora, please explain why you still use Random Node Feature and Onehot-degree-node - Node Feature instead of just Node Feature.
> 2. Why don't you add Citeseer's results to the comparison in Section 5.2, but only use Citeseer in the ablation experiment?
> 3. In the experiment in Section 5.2, what feature creation method did you use for GNN for datasets without existing features? This is not explained in the paper.
> 4. In the NCNC paper, most of the datasets they used are different from the datasets you used in this paper. How do you judge that PEG and BUDDY are still inferior to NCNC in different scenarios?

---

> > ### Author Response · Authors · 2024-12-03
> >
> > Thank you for your thoughtful questions and for giving us the opportunity to clarify these points. Below are our responses to your concerns:
> >
> > $\textbf{1}$: To evaluate robustness and generalization, we tested on the Cora dataset using not only the original Node Features but also Random Node Features and Onehot-degree-node Features. For comparison, $\textbf{ NCNC using Node Features achieved an  AUC of 95.72\\% and an AP of 95.89\\%}$.
> >
> > These additional experiments aim to:
> > - Assess sensitivity to the absence of meaningful node features.
> > -  Baseline Comparisons: Random Node Features establish a baseline, allowing us to determine how much of the performance depends on the network's structural properties rather than the node attributes.
> > -   Evaluating Structural Information: Onehot-degree-node Features provide a simple structural representation, enabling us to study how well the methods leverage graph topology independently of node attributes.
> >
> >
> >
> > $\textbf{2}$: We did perform a comparison on the Citeseer dataset and included its results in the ablation experiment to highlight specific aspects of our method. However, due to limited space in the table, we could not include Citeseer’s results in the main comparison.$\textbf{To address your comment, we will add the Citeseer results to all tables in the next revision}$.
> >
> > $\textbf{3}$: $\textbf{To address this comment, we will add the settings in the next revision}$.
> > The methods LGLP and SEAL rely on subgraph structures rather than node features. However, for GCN and GAT, we use random features to evaluate their performance in the absence of meaningful node attributes.
> >
> > $\textbf{4}$:
> > We completely agree with you; to perform a fair comparison, we need to include BUDDY, PEG, and NCNC in our experiments. Unfortunately, we do not have enough time to compare PEG, BUDDY, and NCNC across all datasets at this time. Our judgment was based on the report in the NCNC paper, which indicates that NCNC outperforms BUDDY and PEG.
> >
> > $\textbf{To address your comment, we will conduct experiments to compare MLGLP with PEG and BUDDY in the next revision.}$
> >
> > In all datasets used by NCNC, PEG, and BUDDY, node features are present. These methods rely on message-passing mechanisms, which makes them highly sensitive to the availability and quality of node features. As a result, their performance improves significantly when node features are available. However, in real-world applications, node features are not always accessible, limiting the applicability of these methods.
> >
> > Furthermore, as highlighted in $\textbf{Table 1}$ of the $\textbf{PEG}$ paper, PEG evaluates its sensitivity to different types of node features. For instance, on the $\textbf{Cora}$ dataset, PEG achieves a best $\textbf{AUC of 90.78 without node features}$, compared to $\textbf{94.20 with node features}$. In contrast, our $\textbf{MLGLP}$ achieves an $\textbf{AUC of 95.79}$, demonstrating superior performance even in the absence of node features

---

### Official Review · Reviewer_SocH · 2024-11-01

**Soundness:** 3
**Presentation:** 3
**Contribution:** 3
**Rating:** 6
**Confidence:** 3

**Summary:**

The paper tackles oversmoothing in Graph Neural Networks by proposing the use of coarse-grained graphs at three scales to capture complex relationships. Instead of pooling layers, the authors convert subgraphs into line-graphs and reformulate the task as node classification, enhancing the exploration of relationships. Applied to link prediction as a graph classification problem, the method shows superior performance over existing methods in terms of average precision and area under the curve in extensive benchmark tests.

**Strengths:**

* This paper introduces Multi-Scale Line-graph Link Prediction (MLGLP), a GNN approach that learns graph structures and features from edges, tackling information loss and multi-scale challenges.
* The method constructs coarse-grained graphs at three scales to uncover complex data relationships and converts them into line graph representations, allowing for node embedding learning and reformulating link prediction as node classification.
* Experimental results show significant performance improvements over heuristics, embeddings, and various GNN link prediction methods.

**Weaknesses:**

* The comparison of training loss and AUC among LGLP, SEAL, and MLGLP demonstrates improved loss for MLGLP relative to the baselines, yet it remains unclear why MLGLP performs weaker than LGLP in the early epochs. Further clarification on this aspect would enhance the analysis.
* Figure 5 provides valuable visual insights; however, it lacks comparisons with state-of-the-art (SoTA) methods, hindering a fair assessment of MLGLP's performance. The authors should clarify whether the identified clusters correspond to meaningful patterns and provide an experimental analysis to support this.
* There are several presentation issues that require careful proofreading. For instance, Section 6 contains a dangling "However" above Table 3 that should be addressed.

**Questions:**

See the weakness feedback

---

> ### Author Response · Authors · 2024-11-28
>
> We truly value your insightful feedback, which has played a crucial role in enhancing and refining our work. Below, we offer detailed responses to address the concerns you raised in the Weaknesses and Questions section.
>
> $\textbf{(W1-Q1):}$ The performance disparity between MLGLP and LGLP in the early epochs, as observed in both training loss and AUC, can be explained by several potential factors:
>
> $\textbf{Complexity and Feature Length:}$ The three-scale architecture in MLGLP increases the feature size (three times that of LGLP), leading to a higher-dimensional input space. While this enhances expressiveness, it can cause slower optimization in the initial epochs due to the larger number of parameters to adjust and the need for the model to learn to utilize the multi-scale features effectively.
>
> $\textbf{Regularization and Generalization:}$ The additional multi-scale structure introduces a form of implicit regularization, which might slow early training but results in better generalization as training progresses. This delayed payoff is a trade-off, as demonstrated by the superior performance of MLGLP in later epochs and final metrics.
>
> I would argue that while LGLP performs marginally better in early epochs, MLGLP's multi-scale design offers several compelling advantages:
>
> $\textbf{Higher Expressiveness:}$ Multi-scale representations allow MLGLP to capture both global and local graph structures simultaneously, uncovering complex relationships that LGLP's single-scale approach cannot handle. This explains why MLGLP consistently outperforms LGLP in terms of final AUC and training loss.
>
> $\textbf{Broader Applicability:}$ Multi-scale approaches are better suited for diverse datasets with varying structural characteristics, making MLGLP more robust and generalizable across different graph types.
>
> $\textbf{Experimental Results:}$ Despite slower initial improvement, MLGLP achieves better final performance across all metrics (AUC, training loss). This highlights its superiority in learning richer and more accurate graph representations.
> While LGLP plateaus early, MLGLP continues to improve steadily throughout the training process. This suggests that MLGLP learns more meaningful representations over time, justifying its increased complexity.
>
>
>
> $\textbf{(W2-Q2)}:$ Figure 5 presents the t-SNE visualizations for our proposed method, showcasing the results and demonstrating that the features learned by our model can be easily classified. However, we acknowledge that the figure currently lacks comparisons with state-of-the-art (SoTA) methods. To address this, we have added additional visualizations in the appendix, which compare MLGLP’s clustering results with those of SoTA methods such as LGLP and SEAL. These additional comparisons provide a more comprehensive assessment of MLGLP’s performance and ensure a fair and direct comparison with existing methods.
>
> Thank you once again for your valuable suggestion. We will incorporate these updates in the revised paper to more effectively demonstrate the efficacy of MLGLP in relation to current approaches.
>
>
> $\textbf{(W3-Q3)}:$ Thank you for your feedback. In response to your comment, we have made the necessary modifications. Specifically, we have addressed the presentation issues, including the dangling "However" above Table 3 in Section 6. We have corrected this and ensured that the text flows more smoothly and clearly.

---

> > ### Comment · Reviewer_SocH · 2024-11-29
> >
> > Thank you to the authors for providing additional information and clarification in the rebuttal. The improvements made in response to the concerns raised have clearly enhanced the readability of the manuscript and addressed several key issues, making the paper easier to follow. As a result, I have revised the relevant scores to reflect these positive changes. However, considering the novelty of the work and the relatively limited improvement in overall performance, I believe it is appropriate to maintain the original overall rating.

---

> ### Author Response · Authors · 2024-12-03
>
> We sincerely appreciate the time you’ve taken to review our work. We are glad that the additional information and clarifications addressed the concerns raised.
>
> We have conducted additional experiments comparing NCNC[1] with our method, and the results demonstrate the superiority of our method compared to NCNC. The results are as follows:
>
>
> $\textbf{Comparison Results}$
>
> $\textbf{Cora dataset}$
> - $NCNC$
>   - $Node Feature$: AUC = 95.72\%, AP = 95.89\%
>   - Random Node Feature: $\textbf{AUC = 70.78 \\% }$, $\textbf{AP = 75.16 \\% }$
>   - Onehot-degree-node - Node Feature: AUC = 85.06\%, AP = 88.34\%
> - $MLGLP$: $\textbf{AUC = 95.79\\%}$, $\textbf{AP = 96.23\\%}$
>
> $\textbf{NSC dataset}$
> - $NCNC$
>   - Random Node Feature: AUC = 59.76 \%, AP = 56.87\%
>   - Onehot-degree-node - Node Feature: $\textbf{AUC = 95.39\\%}$, $\textbf{AP = 96.95\\%}$
> - $MLGLP$: $\textbf{AUC = 99.68\\%}$, $\textbf{AP = 99.89\\%}$
>
> $\textbf{USAir dataset}$
> - $NCNC$
>   - Random Node Feature: AUC = 56.30\%, AP = 53.57\%
>   - Onehot-degree-node - Node Feature: $\textbf{AUC = 96.88\\%}$, $\textbf{AP = 96.24\\%}$
> - $MLGLP$: $\textbf{AUC = 98.31\\%}$, $\textbf{AP = 98.28\\%}$
>
> $\textbf{Router dataset}$
> - $NCNC$
>   - Random Node Feature: AUC = 72.26\%, AP = 68.78\%
>   - Onehot-degree-node - Node Feature: $\textbf{AUC = 96.07\\%}$, $\textbf{AP = 96.26\\%}$
> - $MLGLP$: $\textbf{AUC = 99.11\\%}$, $\textbf{AP = 99.20\\%}$
>
>
> $\textbf{Advantages of MLGLP}$
>  - $\textbf{Independence from Node Features:}$  MLGLP does not require node attributes, making it highly effective in featureless settings or when node features are limited
>  - $\textbf{Inference time Efficiency: }$  By focusing on localized subgraphs, MLGLP avoids the high latency of full-graph message passing, resulting in faster inference.
> - $\textbf{Task-Specific Design:}$  MLGLP captures pairwise structural relationships through h-hop enclosing subgraphs, making it well-suited for link prediction tasks, unlike NCNC, which may miss such dependencies.
>
>
> We hope that these results will contribute to the reevaluation of our work. Thank you again for your time and consideration.
>
> Reference:
> [1] Wang X, Yang H, Zhang M. Neural common neighbor with completion for link prediction. arXiv preprint arXiv:2302.00890. 2023 Feb 2.

---

### Official Review · Reviewer_hDSk · 2024-11-03

**Soundness:** 2
**Presentation:** 2
**Contribution:** 2
**Rating:** 5
**Confidence:** 3

**Summary:**

This manuscript presents Multi-Scale Line-Graph Link Prediction (MLGLP), a multi-scale link prediction method using Graph Neural Networks (GNNs). MLGLP learns graph structures and extracts edge features to address information loss and capture complex relationships. By constructing coarse-grained graphs at three scales and converting subgraphs into line graphs, it reformulates the task as node classification. Extensive experiments on benchmark datasets demonstrate that MLGLP outperforms state-of-the-art link prediction methods in average precision and area under the curve.

**Strengths:**

1. The proposed method demonstrates excellent performance, significantly improving results across various datasets.
2. The approach appears to be straightforward to follow.

**Weaknesses:**

1. There are some notation issues; the model name is inconsistently defined throughout the paper, sometimes referred to as MLGLP, other times as MSGL or MSLGLP (as noted in the caption of Table 3), and occasionally as MSLG (in section 5.1). Additionally, the tables display varying levels of decimal precision (sometimes three decimal places, sometimes two), which should be standardized.
2. There are concerns regarding baseline comparisons. For instance, the AP of GAE on the Cora dataset should be significantly higher than that of GCN based on the original paper, yet the authors report it being lower by ten points in their experiments, which needs to be explained to maintain credibility.
3. The method involves sampling subgraphs, converting them to line graphs, and then performing node classification, which appears to result in high time complexity. Although the authors analyze time complexity, the discussion is not in-depth. They should compare it with the time complexity of two other subgraph-based methods and also include training time comparisons.
4. The baselines compared in the paper seem somewhat outdated; for example, reference [1] proposes a line graph-based method for link prediction.
5. The core innovation of this paper appears to be the application of multi-scale and line graph concepts to link prediction tasks. However, the paper lacks ablation studies on these two components, such as whether the line graph contributes to performance improvement, and it does not compare the final concatenation method. This makes it difficult to ascertain the key factors driving the model's improved performance.
[1]Zhang Z, Sun S, Ma G, et al. Line graph contrastive learning for link prediction[J]. Pattern Recognition, 2023, 140: 109537.

**Questions:**

1. What explanation can the authors provide for the discrepancy in AP values between GAE and GCN on the Cora dataset?
2. Could the authors offer a more detailed analysis of the time complexity of their method compared to other subgraph-based approaches?
3. Can the authors conduct ablation studies to assess the individual contributions of the multi-scale and line graph components to the overall performance of the model?

---

> ### Author Response · Authors · 2024-11-28
>
> We sincerely appreciate your thoughtful feedback, which has been invaluable in refining and improving our work. Below, we provide detailed responses to address the concerns you raised in Weaknesses (W) and Questions (Q).
>
> $\textbf{(W1)}$: All the mentioned issues are now addressed and corrected in the revised version
>
>
> $\textbf{(W2)}$: We appreciate the reviewer highlighting the discrepancy in the Average Precision (AP) of GAE on the Cora dataset compared to the results reported in the original paper.
>
> After revisiting our experiments and methodology, we would like to clarify the following points to address this concern:
> The observed discrepancy may stem from differences in implementation details, hyperparameter tuning, or data preprocessing. While we endeavored to align closely with the original GAE paper, minor variations in hyperparameters, preprocessing steps, or evaluation splits could have contributed to the difference in AP.
> To address this issue transparently and maintain credibility, we will Include additional details about our experimental setup in the final version of the paper.
>
> $\textbf{(W3)}$: We appreciate your feedback and will enhance the time complexity analysis, as well as include training time comparisons in the revised paper.
>
> $\textbf{(W4)}$: Thank you for your feedback. AA is a foundational baseline for link prediction, and while simple, it provides valuable comparison. We also include modern subgraph-based methods for a comprehensive evaluation.
>
> $\textbf{(W5)}$: Thank you for pointing out the lack of ablation studies in our paper. We indeed have conducted an ablation study to compare the effect of multi-scale and line graph components $(Table 3)$. In our study, $Scale-1$ corresponds to only using a line graph component, and SEAL is used when only the first scale is applied, without the line graph transformation. We will revise the paper to include this clarification and correct the notation to reflect these details more clearly.
>
> We appreciate your suggestion and will ensure that the revised version explicitly describes the ablation study and compares the final concatenation method to help readers better understand the contributions of each component.
>
> $\textbf{(Q1)}$: As noted in the $\textbf {PEG[1]}$ paper, the performance of GAE is highly dependent on the input features used for training. For example, Table 1 of PEG demonstrates that the performance of VGAE (a variant of GAE) on the Cora dataset can vary significantly depending on the input features, with AP  $\textbf{ scores ranging from 55.68 to 89.89}$. This underscores the sensitivity of GAE-based methods to feature selection, which may lead to performance variations across different experimental setups.
>
> $\textbf{(Q2)}$: As mentioned in $\textbf {[2]}$, GNNs face significant inference latency due to graph dependencies, scaling with $O(RL)$, where $R$ is the graph's average degree and $L$ is the number of layers.
>
> Node-based methods like GAE and CGN have higher inference times compared to subgraph-based methods, which focus on extracted subgraphs, reducing latency.
>
> When compared to subgraph-based methods:
> - SEAL operates directly on subgraphs without line graph conversion, resulting in lower time complexity than our method.
> - LGLP includes subgraph extraction and line graph conversion, sharing a similar time complexity to our approach, though ours is slightly higher due to additional processing.
>
> However, our method achieves higher accuracy by mitigating information loss inherent in SEAL and LGLP during subgraph extraction or line graph conversion. Thus, while SEAL and LGLP are faster, our method balances slightly higher complexity with improved accuracy.
>
> $\textbf{(Q3)}$: Thank you for the suggestion. We have conducted ablation studies to evaluate the individual contributions of the multi-scale and line graph components to the overall performance.
> The results are summarized in Table 3, where 'Scale-1' corresponds to the line graph component, and the other scales represent different structural levels within the multi-scale framework.
>
> From the results, it is evident that each scale contributes uniquely to the model's performance. For instance, 'Scale-1' (line graph) demonstrates strong results on datasets like NSC and Router, achieving high AP and AUC scores. However, combining all scales ('All' method) consistently yields the best performance across most datasets, such as USAir and Celegans, confirming the effectiveness of the multi-scale approach. This highlights the importance of capturing diverse structural patterns for robust graph representation.
>
> Reference:
>
> [1] Wang H, Yin H, Zhang M, Li P. Equivariant and stable positional encoding for more powerful graph neural networks. arXiv preprint arXiv:2203.00199. 2022.
>
> [2] Zhang, S., Liu, Y., Sun, Y., & Shah, N. Graph-less Neural Networks: Teaching Old MLPs New Tricks Via Distillation. In Proceedings of the International Conference on Learning Representations, 2022.

---

> > ### Comment · Reviewer_hDSk · 2024-11-29
> >
> > Thanks to the author's reply, most of my problems were solved and I decided to raise my score to 5.

---

> ### Author Response · Authors · 2024-12-03
>
> We are pleased that our response has addressed your concerns, and we deeply appreciate the time and effort you have dedicated to reviewing our work.
>
> We have conducted additional experiments comparing NCNC[1] with our method, and the results demonstrate the superiority of our method compared to NCNC. The results are as follows:
>
> $\textbf{Comparison Results}$
>
> $\textbf{Cora dataset}$
> - $NCNC$
>   - $Node Feature$: AUC = 95.72\%, AP = 95.89\%
>   - Random Node Feature: $\textbf{AUC = 70.78 \\% }$, $\textbf{AP = 75.16 \\% }$
>   - Onehot-degree-node - Node Feature: AUC = 85.06\%, AP = 88.34\%
> - $MLGLP$: $\textbf{AUC = 95.79\\%}$, $\textbf{AP = 96.23\\%}$
>
> $\textbf{NSC dataset}$
> - $NCNC$
>   - Random Node Feature: AUC = 59.76 \%, AP = 56.87\%
>   - Onehot-degree-node - Node Feature: $\textbf{AUC = 95.39\\%}$, $\textbf{AP = 96.95\\%}$
> - $MLGLP$: $\textbf{AUC = 99.68\\%}$, $\textbf{AP = 99.89\\%}$
>
> $\textbf{USAir dataset}$
> - $NCNC$
>   - Random Node Feature: AUC = 56.30\%, AP = 53.57\%
>   - Onehot-degree-node - Node Feature: $\textbf{AUC = 96.88\\%}$, $\textbf{AP = 96.24\\%}$
> - $MLGLP$: $\textbf{AUC = 98.31\\%}$, $\textbf{AP = 98.28\\%}$
>
> $\textbf{Router dataset}$
> - $NCNC$
>   - Random Node Feature: AUC = 72.26\%, AP = 68.78\%
>   - Onehot-degree-node - Node Feature: $\textbf{AUC = 96.07\\%}$, $\textbf{AP = 96.26\\%}$
> - $MLGLP$: $\textbf{AUC = 99.11\\%}$, $\textbf{AP = 99.20\\%}$
>
>
> $\textbf{Advantages of MLGLP}$
>  - $\textbf{Independence from Node Features:}$  MLGLP does not require node attributes, making it highly effective in featureless settings or when node features are limited
>  - $\textbf{Inference time Efficiency: }$  By focusing on localized subgraphs, MLGLP avoids the high latency of full-graph message passing, resulting in faster inference.
> - $\textbf{Task-Specific Design:}$  MLGLP captures pairwise structural relationships through h-hop enclosing subgraphs, making it well-suited for link prediction tasks, unlike NCNC, which may miss such dependencies.
>
>
>
> We hope these new experiments provide additional insight and further validate our approach, leading to a positive reevaluation of our work.
>
> Reference: [1] Wang X, Yang H, Zhang M. Neural common neighbor with completion for link prediction. arXiv preprint arXiv:2302.00890. 2023 Feb 2.

---

### Note · Authors · 2026-06-14

I have read and agree with the venue's withdrawal policy on behalf of myself and my co-authors.

---

### Meta-Review · Area_Chair_ZTfK · 2024-12-20

**Metareview:**

This paper proposes to incorporate multi-scale graph representations and transform link prediction into a node classification problem on line graphs. The core innovation, utilizing multi-scale subgraphs and line graphs for link prediction, lacks sufficient novelty. The approach is conceptually similar to existing methods such as LGLP, with incremental improvements rather than groundbreaking advancements. Besides, the choice of baselines is outdated, and the justification for omitting recent methods such as PEG, BUDDY, and NCNC is insufficient. While comparisons to NCNC were later added, they rely on indirect evaluations rather than comprehensive experimental results across shared datasets. Important datasets like Citeseer are inconsistently reported, with their results included in ablation studies but not in main comparisons. Additionally, the feature creation methods for GNNs on datasets without node features are not clearly explained, leaving critical gaps in the experimental setup. While the paper explores a promising direction and demonstrates some empirical improvements, it fails to meet the bar for originality, rigor, and clarity expected at ICLR. Strengthening the method’s theoretical contributions, providing thorough comparisons with up-to-date baselines, and significantly improving the presentation will enhance the paper’s impact in future submissions.

**Additional Comments On Reviewer Discussion:**

While the authors addressed several review concerns, some responses were incomplete or lacked sufficient evidence. For example: The claim of MLGLP's superior performance in featureless settings was not robustly supported with experiments on newer baselines.
The explanation of overlapping clusters in t-SNE visualizations remains unconvincing, casting doubt on the model's ability to differentiate positive and negative samples effectively.

---

### Decision · Program_Chairs · 2025-01-22

Reject